# Interventions, Where and How?
# Experimental Design for Causal Models at Scale

**Panagiotis Tigas**[*1]   **Yashas Annadani**[*2,3]   **Andrew Jesson**[1]   **Bernhard Schölkopf**[3]
**Yarin Gal**[1]   **Stefan Bauer**[2,4]

[1]OATML, University of Oxford   [2]KTH Royal Institute of Technology, Stockholm
[3]Max Planck Institute for Intelligent Systems   [4]CIFAR Azrieli Global Scholar

## Abstract

Causal discovery from observational and interventional data is challenging due to limited data and non-identifiability: factors that introduce uncertainty in estimating the underlying structural causal model (SCM). Selecting experiments (interventions) based on the uncertainty arising from both factors can expedite the identification of the SCM. Existing methods in experimental design for causal discovery from limited data either rely on linear assumptions for the SCM or select only the intervention target. This work incorporates recent advances in Bayesian causal discovery into the Bayesian optimal experimental design framework, allowing for active causal discovery of large, nonlinear SCMs while selecting both the interventional target and the value. We demonstrate the performance of the proposed method on synthetic graphs (Erdos-Rènyi, Scale Free) for both linear and nonlinear SCMs as well as on the *in-silico* single-cell gene regulatory network dataset, DREAM.

## 1   Introduction

What is the structure of the protein-signaling network derived from a single cell? How do different habits influence the presence of disease? Such questions refer to causal effects in complex systems governed by nonlinear, noisy processes. On most occasions, passive observation of such systems is insufficient to uncover the real cause-effect relationship and costly experimentation is required to disambiguate between competing hypotheses. As such, the design of experiments is of significant interest; an efficient experimentation protocol helps reduce the costs involved in experimentation while aiding the process of producing knowledge through the (closed-loop / policy-driven) scientific method (Fig. 1).

In the language of causality (Pearl, 2009), the causal relationships are represented qualitatively by a directed acyclic graph (DAG), where the nodes correspond to different variables of the system of study and the edges represent the flow of information between the variables. The abstraction of DAGs allows us to represent the space of possible explanations (hypotheses) for the observations at hand. Representing such hypotheses as Bayesian probabilities (beliefs) allows us to formalize the problem of the scientific method as one of

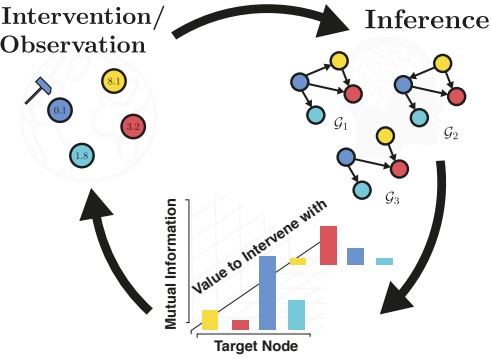

**Figure 1:** Intervention-Inference-Design loop of Bayesian Optimal Experimental Design for Causal Discovery framework.

---

*Equal contribution. Correspondence to ptigas@robots.ox.ac.uk, yashas.annadani@gmail.com.
Implementation available at: https://github.com/yannadani/cbed

Bayesian inference, where the goal is to estimate the posterior distribution $p(\text{DAGs} \mid \text{Observations})$. A posterior distribution over the DAGs allows us to employ information-theoretic acquisition functions that guide experimentation towards the most informative variables for disambiguating between competing hypotheses. Such design procedures belong to the field of *Bayesian Optimal Experimental Design* (Lindley, 1956) for *Causal Discovery* (BOECD) (Tong and Koller, 2001, Murphy, 2001).

In the *Bayesian Optimal Experimental Design* (BOED) (Lindley, 1956) framework, one seeks the experiment that maximizes the *expected information gain* about some parameter(s) of interest. In *causal discovery*, an experiment takes the form of a causal intervention, and the parameters of interest are the Structural Causal Model (SCM) and its associated DAG.

An intervention in a causal model refers to the variable (or target) we manipulate and the value (or strength) at which we set the variable. Hence, the design space in the case of learning causal models is the set of all subsets of the intervention targets and the possibly countably infinite set of intervention values of the chosen targets. The intervention value encapsulates important semantics in many causal inference applications. For instance, in medical applications, an intervention can correspond to the administration of different drugs and the intervention value takes the form of a dosage level for each drug. Even though the appropriate choice of this value is crucial for identifying the underlying causal model, existing work on active causal discovery focuses exclusively on selecting the intervention target (Agrawal et al., 2019, Cho et al., 2016). There, the intervention value is generally some arbitrary fixed value (like 0) which is suboptimal (see Fig. 2a). Hence, a holistic treatment of selecting the intervention value and the target in the general case of nonlinear causal models has been missing. We present a Bayesian experimental design method (CBED - pronounced "seabed") to acquire optimal intervention targets and values by performing Bayesian optimization.

Additionally, some settings call for the selection of a batch of interventions. The problem of batched interventions is computationally expensive as it requires evaluating all possible combinations of interventions. We extend CBED to the batch setting and propose two different batching strategies for tractable, Bayes optimal acquisition of both intervention targets and values. The first strategy — Greedy-CBED — builds up the intervention set greedily. A greedy heuristic is still near-optimal due to submodularity properties of mutual information (Krause and Guestrin, 2012, Agrawal et al., 2019, Kirsch et al., 2019). The second strategy — Soft-CBED — constructs a set of interventions by stochastic sampling from a finite set of candidates, thereby significantly increasing computational efficiency while recovering the DAG structure and the parameters of the SCM as fast as the greedy strategy. This strategy is well suited for resource-constrained settings.

Throughout this work, we make the following standard assumptions for causal discovery (Peters et al., 2017):

**Assumption 1** (Causal Sufficiency). *There are no hidden confounders, and all the random variables of interest are observable.*

**Assumption 2** (Finite Samples). *There is a finite number of observational/ interventional samples available.*

**Assumption 3** (Nonlinear SCM with Additive Noise). *The structural causal model has nonlinear conditional expectations with additive Gaussian noise.*

**Assumption 4** (Single Target). *Each intervention is atomic and applied to a single target of the SCM.*

Additionally, we assume that interventions are planned and executed in batches of size $\mathcal{B}$, with a fixed budget of total interventions given by Number of Batches $\times \mathcal{B}$. We also assume that the underlying graph is sparse, as is the case in all the real-world settings (Bengio et al., 2019, Schmidt et al., 2007). Experimental design is preferable in sparse graph settings as the number of informative intervention targets and values would be significantly less compared to dense graphs. Many nodes corresponding to a sparse graph would have very less probability of having parent sets, and hence preforming experiments with a random policy is not maximally informative. Finally, we are interested in recovering the full graph $\mathbf{G}$ with a small number of batches. As with all causal inference tasks, the assumptions that we make above have to be carefully verified for the application of interest.

We show that our methods, Greedy-CBED and Soft-CBED, perform better than the state-of-the-art active causal discovery baselines in linear and nonlinear SCM settings. In addition, our approach achieves superior results in the real-world inspired nonlinear dataset, DREAM (Greenfield et al., 2010).

## 2 Background

**Notation.** Let $\mathbf{V} = \{1, \ldots, d\}$ be the vertex set of any DAG $\mathbf{g} = (\mathbf{V}, E)$ and $\mathbf{X}_{\mathbf{V}} = \{X_1, \ldots, X_d\} \subseteq \mathcal{X}$ be the random variables of interest indexed by $\mathbf{V}$. We have an initial observational dataset $\mathcal{D} = \{\mathbf{x}_{\mathbf{V}}^{(i)}\}_{i=1}^n$ comprised of instances $\mathbf{x}_{\mathbf{V}} \sim P(X_1 = x_1, \ldots, X_d = x_d) = p(x_1, \ldots, x_d)$.

**Causal Bayesian Network.** A causal Bayesian network (CBN) is the pair $(\mathbf{g}, P)$ such that for any $\mathbf{W} \subset \mathbf{V}$,

$$P(\mathbf{X}_{\mathbf{V}} | \mathrm{do}(\mathbf{X}_{\mathbf{W}} = \mathbf{x}'_{\mathbf{W}})) = \prod_{i \in \mathbf{V} \setminus \mathbf{W}} P(X_i \mid X_{\mathrm{pa}_{\mathbf{g}}(i)}) \mathbb{1}(\mathbf{X}_{\mathbf{W}} = \mathbf{x}'_{\mathbf{W}})$$

where $\mathrm{do}(X_{\mathbf{W}})$ represents a hypothetical intervention on the variables $X_{\mathbf{W}}$, $\mathbb{1}(\cdot)$ is an indicator function and $\mathrm{pa}_{\mathbf{g}}(i)$ denotes parents of variable $X_i$ in DAG $\mathbf{g}$. A perfect intervention on any variable $X_j$ completely removes all dependencies with its parents, i.e. $P(X_j \mid X_{\mathrm{pa}_{\mathbf{g}}(j)}) = P(X_j)$ thereby resulting in a mutilated DAG $\mathbf{g}' = (\mathbf{V}, E \setminus (pa_{\mathbf{g}}(j), j))$.

**Structural Causal Model.** From the data generative mechanism point of view, the DAG $\mathbf{g}$ on $\mathbf{X}_{\mathbf{V}}$ matches a set of *structural equations*:

$$X_i := f_i(X_{\mathrm{pa}_{\mathbf{g}}(i)}, \epsilon_i) \quad \forall i \in \mathbf{V} \tag{1}$$

where $f_i$'s are (potentially nonlinear) causal mechanisms that remain invariant when intervening on any variable $X_j \neq X_i$. $\epsilon_i$'s are exogenous noise variables with arbitrary distribution that are mutually independent, i.e $\epsilon_i \perp\!\!\!\perp \epsilon_j \forall i \neq j$. (1) represents the conditional distributions in a Causal Bayesian Network and can additionally reveal the effect of interventions if the mechanisms are known (Peters et al., 2017, Pearl, 2009). These equations together form the structural causal model (SCM), with an associated DAG $\mathbf{g}$. Though the mechanisms $f$ can be nonparametric in the general case, we assume that there exists a parametric approximation to these mechanisms with parameters $\boldsymbol{\gamma} \in \Gamma$. In the case of linear SCMs, $\boldsymbol{\gamma}$ corresponds to the weights of the edges in $E$. In the nonlinear case, they could represent the parameters of a nonlinear function that parameterizes the mean of a Gaussian distribution.

A common form of (1) corresponds to Gaussian additive noise models (ANM)[2]:

$$X_i := f_i(X_{\mathrm{pa}_{\mathbf{g}}(i)}; \boldsymbol{\gamma}_i) + \epsilon_i, \quad \epsilon_i \sim \mathcal{N}(0, \sigma_i^2) \tag{2}$$

An ANM is fully specified by a a DAG $\mathbf{g}$, mechanisms, $f(\cdot; \boldsymbol{\gamma}) = [f_1(\cdot; \gamma_1), \ldots, f_d(\cdot; \gamma_d)]$, parameterized by $\boldsymbol{\gamma} = [\gamma_1, \ldots, \gamma_d]$, and variances, $\sigma^2 = [\sigma_1^2, \ldots, \sigma_d^2]$. For notational brevity, henceforth we denote $\boldsymbol{\theta} = (\boldsymbol{\gamma}, \sigma^2)$ and all the parameters of interest with $\boldsymbol{\phi} = (\mathbf{g}, \boldsymbol{\theta})$.

**Bayesian Causal Discovery.** A common assumption in causal inference is that causal relations are known qualitatively and can be represented by a DAG. While this qualitative information can be obtained from domain knowledge in some scenarios, it's infeasible in most applications. The goal of causal discovery is to recover the SCM and the associated DAG, given a dataset $\mathcal{D}$. In general, without further assumptions about the nature of mechanisms $f$ (e.g., linear vs. nonlinear), the true SCM may not be *identifiable* (Peters et al., 2012) from observational data alone. This non-identifiability is because there could be multiple DAGs (and hence multiple factorizations of $P(\mathbf{X}_{\mathbf{V}})$) which explain the data equally well. Such DAGs are said to be *Markov Equivalent*. Interventions can improve identifiability. In addition to identifiability issues, estimating the functional relationships between nodes using finite data is another source of uncertainty. Bayesian parameter estimation over the unknown SCM provides a principled way to quantify these uncertainties and obtain a posterior distribution over the SCM given observational data. An experimenter can then use the knowledge encoded by the posterior to design informative experiments that efficiently *acquire interventional data to resolve unknown edge orientations and functional uncertainty.*

---

[2]ANM's can have noise variables that are non-Gaussian as well, but we restrict our exposition to the Gaussian case.

**Bayesian Inference of SCMs and DAGs.** The key challenge in performing Bayesian inference jointly over SCMs and DAGs is that the space of DAGs is discrete and superexponential in the number of variables (Peters et al., 2017). However, recent techniques based on variational inference (Annadani et al., 2021, Lorch et al., 2021, Cundy et al., 2021) provide a tractable and scalable way of performing posterior inference of these parameters. Given a tractable distribution $q_\psi(\phi)$ which approximates the posterior $p(\phi \mid \mathcal{D})$, variational inference maximizes a lower bound on the (log-) evidence:

$$\log p(\mathcal{D}) \geq \mathcal{L}(\psi \in \Psi) = \mathop{\mathbb{E}}_{q_\psi(\phi)} \left[ \log p(\mathcal{D} \mid \phi) \right] - D_{\mathrm{KL}}(q_\psi(\phi) || p(\phi))$$

The key idea in these techniques is the way the variational family $\Psi$ for DAGs is parameterized. The variational family for the Variational Causal Network (VCN) method (Annadani et al., 2021) is an autoregressive Bernoulli distribution over the adjacency matrix. They further enforce the acyclicity constraint (Zheng et al., 2018) through the prior. BCD-Nets (Cundy et al., 2021) consider a distribution over node orderings through a Boltzmann distribution and perform inference with Gumbel-Sinkhorn (Mena et al., 2018) operator. DiBS (Lorch et al., 2021) consider latent variables over entries of adjacency matrix and perform inference over these latent variables using SVGD (Liu and Wang, 2016). We demonstrate empirical results in BOECD using the DiBS model in this work because it is easily extendable to nonlinear SCMs.

**Bayesian Optimal Experimental Design.** *Bayesian Optimal Experimental Design* (BOED) (Lindley, 1956, Chaloner and Verdinelli, 1995) is an information theoretic approach to the problem of selecting the optimal experiment to estimate any parameter $\theta$. For BOED, the *utility* of the experiment $\xi$ is the mutual information (MI) between the observation $\mathbf{y}$ and $\theta$:

$$U_{\mathrm{BOED}}(\xi) \triangleq I(\mathbf{Y}; \theta \mid \xi, \mathcal{D}) = \mathop{\mathbb{E}}_{p(\mathbf{y}|\theta,\xi)p(\theta|\mathcal{D})} \left[ \log p(\mathbf{y} \mid \xi, \mathcal{D}) - \log p(\mathbf{y} \mid \theta, \xi, \mathcal{D}) \right]$$

The goal of BOED is to select the experiment that maximizes this objective $\xi^* = \arg\max_\xi U_{\mathrm{BOED}}(\xi)$. Unfortunately, evaluating and optimizing this objective is challenging because of the nested expectations (Rainforth et al., 2018) and several estimators have been introduced (Foster et al., 2019, Kleinegesse and Gutmann, 2019) that lower bound the BOED objective which then can be combined with various optimization methods to select the designs (Foster et al., 2020, Ivanova et al., 2021, Foster et al., 2021, Lim et al., 2022).

A common setting, called *static*, *fixed* or *batch* design, is to optimize $\mathcal{B}$ designs $\{\xi_1, \ldots, \xi_\mathcal{B}\}$ at the same time. The designs are then executed and the experimental outcomes are collected to update the model parameters in a Bayesian fashion.

## 3   Method

The true SCM and the associated DAG $\tilde{\phi} = (\tilde{\mathbf{g}}, \tilde{\theta})$ over random variables $\mathbf{X_V}$ is a matter of fact, but our belief in $\tilde{\phi}$ is uncertain for many reasons. Primarily, it is only possible to learn the DAG $\tilde{\mathbf{g}}$ up to a Markov equivalence class (MEC) from observational data $\mathcal{D}$. Uncertainty also arises from $\mathcal{D}$ from being a finite sample, which we model by introducing the the random variable $\mathbf{\Phi}$, of which $\phi$ is an outcome. Let $\phi \sim p(\phi|\mathcal{D}) \propto p(\mathcal{D} \mid \phi)p(\phi)$ be an instance of the random variable $\mathbf{\Phi}$ that is sampled from our posterior over SCMs after observing the dataset $\mathcal{D}$.

We would like to design an experiment to identify an intervention $\xi := \{(j, v)\} := \mathrm{do}(X_j = v)$ that maximizes the information gain about $\mathbf{\Phi}$ after observing the outcome of the intervention $\mathbf{y} \sim P(X_1 = x_1, \ldots, X_d = x_d \mid \mathrm{do}(X_j = v) = p(\mathbf{y} \mid \xi)$. Here, $\mathbf{y}$ is an instance of the random variable $\mathbf{Y} \subseteq \mathcal{X}$ distributed according to the distribution specified by the mutilated true graph $\tilde{\mathbf{g}}'$ under intervention. Looking at one intervention at a time, one can formalize BOECD as gain in information about $\mathbf{\Phi}$ after observing the outcome of an experiment $\mathbf{y}$. The experiment $\xi := \{(j, v)\}$ that maximizes the information gain is the experiment that maximizes the mutual information between $\mathbf{\Phi}$ and $\mathbf{Y}$:

$$\{(j^*, v^*)\} = \arg\max_{j,v} \{I(\mathbf{Y}; \mathbf{\Phi} \mid \{(j, v)\}, \mathcal{D})\} \tag{3}$$

The above objective considers taking `arg max` over not just the discrete set of intervention targets $j \in \mathbf{V}$, but also over the uncountable set of intervention values $v \subset \mathcal{X}_j$. While the existing works in BOECD consider only the design of intervention targets to limit the complexity (Tong and Koller, 2001, Murphy, 2001, Agrawal et al., 2019), our approach tackles both the problems. We first outline the methodology for a single design and in Section 3.2 demonstrate how to extend this single design to a batch setting.

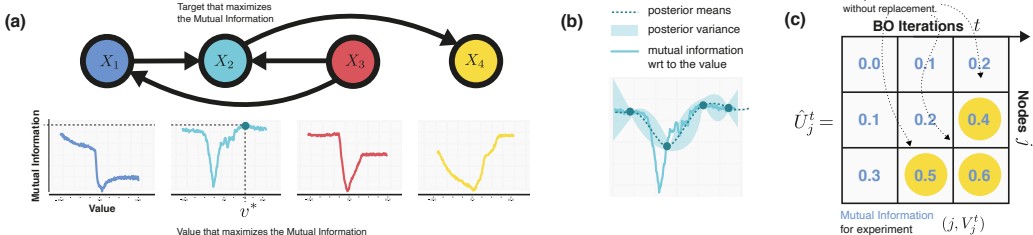

**Figure 2: (a)** Each graph shows how the **mutual information** (MI) (y-axis) changes for intervening on that node (plot color matching the node color) with different **values** (x-axis). The SCM in this example is a nonlinear SCM with Additive Gaussian noise. We can see that by intervening on node $X_2$ with the value $v^*$, the mutual information gets maximized. **(b)** The posterior distribution of a GP on the Mutual Information function as a response to different intervention values after four Bayesian Optimization (BO) steps. **(c)** For each $t$ iteration of the BO algorithm and each node $j$, we get a utility function evaluation $\hat{U}_j^t$ (the utility being the MI in our case). Then we sample without replacement proportionally to the scores to prepare a batch (3.2).

### 3.1 Single Design

To maximize the objective in Equation 3, we need to (1) estimate MI for candidate interventions and (2) maximize the estimated MI by optimizing over the domain of intervention value for every candidate interventional target.

**Estimating the MI.** As mutual information is intractable, there are various ways to estimate it depending on whether we can sample from the posterior and whether the likelihood can be evaluated (Foster et al., 2020, Poole et al., 2019, Houlsby et al., 2011). Since the models we consider allow both posterior sampling and likelihood evaluation, it suffices to obtain an estimator which requires only likelihood evaluation and Monte Carlo approximations of the expectations. To do so, we derive an estimator similar to Bayesian Active Learning by Disagreement (BALD) (Houlsby et al., 2011), which considers MI as a difference of conditional entropies over the outcomes $\mathbf{Y}$:

$$\mathrm{I}(\mathbf{Y}; \mathbf{\Phi} \mid \{(j,v)\}, \mathcal{D}) = \mathrm{H}(\mathbf{Y} \mid \{(j,v)\}, \mathcal{D}) - \mathrm{H}(\mathbf{Y} \mid \mathbf{\Phi}, \{(j,v)\}, \mathcal{D})$$

$$= -\mathop{\mathbb{E}}_{p(\mathbf{y}\mid\{(j,v)\}, \mathcal{D})}\left[\log\left(\mathop{\mathbb{E}}_{p(\phi\mid\mathcal{D})}[p(\mathbf{y} \mid \phi, \{(j,v)\})]\right)\right] + \mathop{\mathbb{E}}_{p(\phi\mid\mathcal{D})}\left[\mathop{\mathbb{E}}_{p(\mathbf{y}\mid\phi,\{(j,v)\})}[\log\left(p(\mathbf{y} \mid \phi, \{(j,v)\})\right)]\right] \quad (4)$$

where $\mathrm{H}(\cdot)$ is the entropy. See Appendix B.1 for the derivation. A Monte Carlo estimator of the above equation can be used as an approximation (Appendix B.2). Equation (4) has an intuitive interpretation. It assigns high mutual information to interventions that the model disagrees the most regarding the outcome. We denote the MI for a single design as $\mathcal{I}(\{(j,v)\}) \coloneqq \mathrm{I}(\mathbf{Y}; \mathbf{\Phi} \mid \{(j,v)\}, \mathcal{D})$.

**Selecting the Intervention Value.** As shown in (3), maximizing the objective is achieved not only by selecting the intervention target but also by setting the appropriate intervention value. Although optimizing the intervention target is tractable (discrete and finite number of nodes to select from), selecting the value to intervene is usually intractable since they are continuous. For any given target node $j$, MI is a nonlinear function over $v \in \mathcal{X}_j$ (See Fig 2) and hence solving with gradient ascent techniques only yields a local maximum. Given that MI is expensive to evaluate, we treat MI for a given target node $j$ as a black-box function and obtain its maximum using Bayesian Optimization (BO) (Kushner, 1964, Zhilinskas, 1975, Močkus, 1975). BO seeks to find the maximum of this function $max_{v \in \mathcal{X}_j}\mathcal{I}(\{(j,v)\})$ over the entire set $\mathcal{X}_j$ with as few evaluations as possible. See appendix E for details.

BO typically proceeds by placing a Gaussian Process (GP) (Rasmussen, 2003) prior on the function $\mathcal{I}(\{j, \cdot\})$ and obtain the posterior of this function with the queried points $\mathbf{v}^* = \{v^{(1)*}, \ldots, v^{(T)*}\}$. Let the value of the mutual information queried at each optimization step $t$ be $\hat{\mathrm{U}}_j^t = \mathcal{I}(\{(j, v^{(t)*})\})$. The posterior *predictive* of a point $v^{(t+1)}$ can be obtained in closed form as a Gaussian with mean $\boldsymbol{\mu}_j^{(t+1)}(v)$ and variance $\boldsymbol{\sigma}_j^{(t+1)}(v)$. Subscript $j$ signifies the fact that we maintain different $\mu$ and $\sigma$ per intervention target and superscript $t$ represents the BO step. Querying proceeds by having an acquisition function defined on this posterior, which suggests the next point to query. For BO, we use

an acquisition function called as the Upper Confidence Bound (UCB) (Srinivas et al., 2010) which suggests the next point to query by trading-off *exploration* and *exploitation* with a hyperparameter $\beta_j^t$: $v_j^{(t+1)*} = \arg\max_v \boldsymbol{\mu}_j^t(v) + \sqrt{\beta_j^{t+1}} \boldsymbol{\sigma}_j^t(v)$.

We run GP-UCB independently on every candidate intervention target $j = \{1, \ldots, d\}$ by querying points within a fixed domain $[-k, k] \subset \mathbb{R}$. Note that the domain can be chosen based on the application, for example, if we must constrain dosage levels within a fixed range. Each GP is one-dimensional in our setup; hence a few evaluations of UCB are sufficient to get a good value maxima candidate. Further, GP-UCB for each candidate target is parallelizable, making it efficient. We finally select the design with the highest MI across the candidate intervention targets.

## 3.2 Batch Design

In many applications, it is desirable to select the most informative *set* of interventions instead of a single intervention at a time. Take, for example, a biologist entering a wet lab with a script of experiments to execute. Batching experiments removes the bottleneck of waiting for an experiment to finish and get analyzed until executing the next one. Given a budget per batch $\mathcal{B}$ which denotes the number of experiments in a batch, the problem of selecting the batch then becomes $\arg\max_{\boldsymbol{\Xi}} \mathrm{I}(\mathbf{Y}; \boldsymbol{\Phi} \mid \boldsymbol{\Xi}, \mathcal{D})$, such that $\texttt{cardinality}(\boldsymbol{\Xi}) = \mathcal{B}$, where $\boldsymbol{\Xi}$ is a set of interventions $\bigcup_{i=1}^{\mathcal{B}} (j_i, v_i)$ and $\mathbf{Y}$ denotes the random variable for the outcomes of the interventions of the batch. We denote the MI for a batch design as $\mathcal{I}(\boldsymbol{\Xi}) \coloneqq \mathrm{I}(\mathbf{Y}; \boldsymbol{\Phi} \mid \boldsymbol{\Xi}, \mathcal{D})$.

**Greedy Algorithm.** Computing the optimal solution $\mathcal{I}(\boldsymbol{\Xi}^*)$ is computationally infeasible. However, as the conditional mutual information is *submodular* and *non-decreasing* (see Appendix B.4 for proof), we can derive a simple greedy algorithm (Algorithm 1) that can achieve at least a $(1 - 1/e) \approx 0.64$ approximation of the optimal solution (Krause and Guestrin, 2012, Nemhauser et al., 1978). We denote this strategy as `Greedy-CBED`.

**Soft Top-K.** Although the greedy algorithm is tractable, it requires $O(\mathcal{B}d)$ instances of GP-UCB. Kirsch et al. (2021) show that a soft top-k selection strategy performs similarly to the greedy algorithm, reducing the computation requirements to $O(d)$ runs of GP-UCB. To achieve this, we construct a finite set of candidate intervention target-value pairs by keeping all the $T$ evaluations of GP-UCB for each node $j = \{1, \ldots, d\}$. Therefore, for $d$ nodes, our candidate set is comprised of $d \times T$ experiments. We score each experiment in this candidate set using the MI estimate. We then sample *without replacement* $\mathcal{B}$ times proportionally to the *softmax* of the MI scores (Algorithm 2). We denote this strategy as `Soft-CBED`.

## 3.3 Comparison with existing active causal discovery methods

We outline how our approach compares with two main existing active causal discovery methods.

**ABCD (Agrawal et al., 2019).** The estimator of MI used in ABCD is based on weighted importance sampling. However, for the specific choice of the importance sampling weights used in ABCD, their MI estimator ends up with the same approximation as in our method (see Appendix B.5). Nevertheless, ABCD does not select intervention values but suboptimally sets them to a fixed value. In addition, our proposed `Soft-CBED` is a faster and more efficient batch strategy, especially when values also have to be acquired. From this perspective, our approach is an extension of ABCD with nonlinear assumptions, value acquisition, and a soft top-k batching strategy.

**AIT (Scherrer et al., 2021).** AIT is an F-score-based intervention target acquisition strategy. Although it is not a BOECD method, we prove here that it can be viewed as a Monte Carlo estimate of the approximation to MI when the outcomes $\mathbf{Y}$ are Gaussian. Nevertheless, AIT does not select intervention values like ABCD and does not have a batch strategy.

**Theorem 3.1.** *Let* $\mathbf{Y}$ *be a Gaussian random variable. Then the discrepancy score of Scherrer et al. (2021) is a Monte Carlo estimate of an approximation to mutual information (Eq. (4)). See Appendix B.6 for proof.*

| **Algorithm 1:** `Greedy-CBED` | **Algorithm 2:** `Soft-CBED` |
|---|---|

**Algorithm 1: Greedy-CBED**

**Input** : $\mathcal{E}$ environment, $N$ initial observational samples, $\mathcal{B}$ batch Size, $d$ number of nodes

▷ Initialize set of experiments $\Xi$ to empty

1   $\Xi \leftarrow \varnothing$
2   **for** $n = 1 \ldots \mathcal{B}$ **do**
3     **for** $j = 1 \ldots d$ **do**
      ▷ Select optimal intervention value per node $j$ using GP-UCB
4       $V_j \leftarrow \arg\max_v \mathcal{I}(\Xi \cup \{(j, v)\})$
5       $U_j \leftarrow \mathcal{I}(\Xi \cup \{(j, V_j)\})$
6     $j^* \leftarrow \arg\max_j U_j$
7     $v^* \leftarrow V_{j^*}$
8     $\Xi \leftarrow \Xi \cup \{(j^*, v^*)\}$
9   **return** $\Xi$

**Algorithm 2: Soft-CBED**

**Input** : $\mathcal{E}$ environment, $N$ initial observational samples, $\mathcal{B}$ batch Size, $d$ number of nodes, $\zeta$ softmax temperature

1   **for** $j = 1 \ldots d$ **do**
    ▷ Select candidate intervention values per node $j$ using GP-UCB
2     Initialize $\mu_j^0$ and $\sigma_j^0$
3     **for** $t = 1 \ldots T$ **do**
4       $V_j^t \leftarrow \arg\max_v \mu_j^{t-1}(v) + \sqrt{\beta^t}\sigma_j^{t-1}(v)$
5       $\hat{U}_j^t \leftarrow \mathcal{I}(\{(j, V_j^t)\})$
6       Update the GP to obtain $\mu_j^t$ and $\sigma_j^t$
7   $\{(t_i, j_i)\}_{i \in \{1, \ldots, \mathcal{B}\}} \leftarrow \mathcal{B}$ samples *without* replacement $\propto \exp(\hat{U}_j^t / \zeta)$
8   $\Xi \leftarrow \{(j_i, V_{j_i}^{t_i})\}_{i \in \{1, \ldots, \mathcal{B}\}}$
9   **return** $\Xi$

# 4   Related Work

Early efforts of using *Bayesian Optimal Experimental Design for Causal Discovery* (BOECD) can be found in the works of Murphy (2001) and Tong and Koller (2001). However, these approaches deal with simple settings like limiting the graphs to topologically ordered structures, intervening sequentially, linear models, and discrete variables.

In Cho et al. (2016) and Ness et al. (2017), BOECD was applied for learning biological networks structure. BOECD was also explored in Greenewald et al. (2019) under the assumption that undirected edges of the graph always forms a tree. More recently, ABCD framework (Agrawal et al., 2019) extended the work of Murphy (2001) and Tong and Koller (2001) in the setting where interventions can be applied in batches with continuous variables. To achieve this, they (approximately) solve the submodular problem of maximizing the batched mutual information between interventions (experiments), outcomes, and observational data, given a DAG. DAG hypotheses are sampled using *DAG-bootstrap* (Friedman et al., 2013). Our work differs from ABCD in a few ways: we work with both linear and nonlinear SCMs by using state-of-the-art posterior models over DAGs (Lorch et al., 2021), we apply BO to select the value to intervene with, but we also prepare the batch using *soft*BALD (Kirsch et al., 2021) which is significantly faster than the greedy approximation of ABCD method.

In von Kügelgen et al. (2019) the authors proposed the use of *Gaussian Processes* to model the posterior over DAGs and then use BO to identify the value to intervene with, however, this method was not shown to be scalable for larger than bivariate graphs since they rely on multi-dimensional Gaussian Processes for modeling the conditional distributions.

A new body of work has emerged in the field of differentiable causal discovery, where the problem of finding the structure, usually from observational data, is solved with gradient ascent and functional approximators, like neural networks (Zheng et al., 2018, Ke et al., 2019, Brouillard et al., 2020, Bengio et al., 2019). In recent works (Cundy et al., 2021, Lorch et al., 2021, Annadani et al., 2021), the authors proposed a variational approximation of the posterior over the DAGs which allowed for modeling a distribution rather than a point estimate of the DAG that best explains the observational data $\mathcal{D}$. Such work can be used to replace *DAG-bootstrap* (Friedman et al., 2013), allowing for the modeling of posterior distributions with greater support.

Besides the BOECD-based approaches, a few active causal learning works have been proposed (He and Geng, 2008, Gamella and Heinze-Deml, 2020, Scherrer et al., 2021, Shanmugam et al., 2015, Squires et al., 2020, Kocaoglu et al., 2017). Active ICP (Gamella and Heinze-Deml, 2020) uses ICP (Peters et al., 2016) for causal learning while using an active policy to select the target, however,

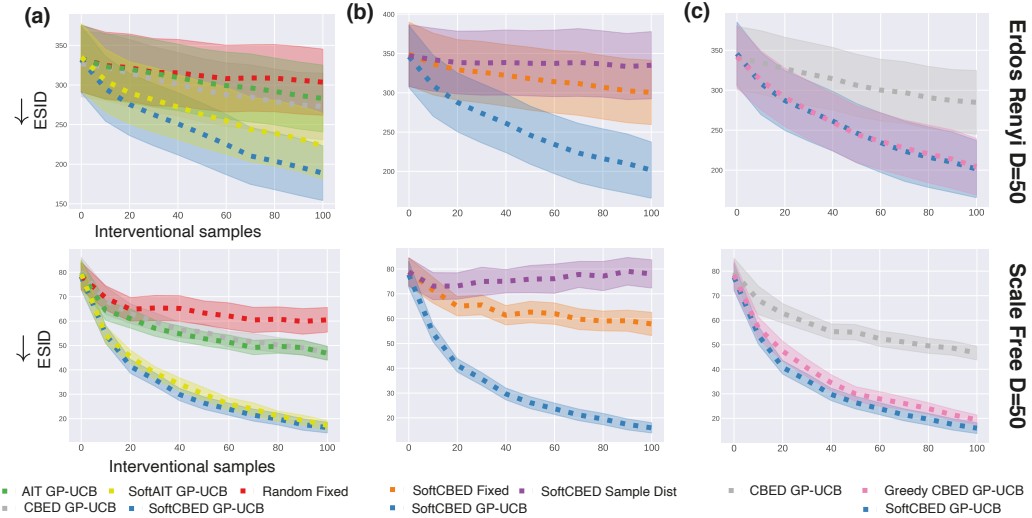

**Figure 3:** Results on the $\mathbb{E}$-**SID** $\downarrow$ metric (100 seeds, with standard error of the mean shaded) for 50 variables involving nonlinear functional relationships and additive Gaussian noise. (a) We show that `Soft-CBED` with GP-UCB value selection strategy significantly outperforms the baselines. (b) We isolate the effect of the value selection strategy. We show that intervening with a fixed value and sampling from the support of data both perform worse than having an optimizer like GP-UCB. (c) we compare non-batch (CBED) vs batch-based acquisition functions (`Greedy-CBED`, `Soft-CBED`). As we can see, `Soft-CBED` performs as well as `Greedy-CBED`. For all experiments, we use the DiBS (Lorch et al., 2021) posterior model.

this work is not applicable in the setting where the full graph needs to be recovered. In Zhang et al. (2021), the authors propose an active learning method to the problem of identifying the interventions that push a dynamical causal network towards a desired state. A few approaches tackle the problem of actively acquiring interventional data to orient edges of a skeletal graph (Shanmugam et al., 2015, Squires et al., 2020, Kocaoglu et al., 2017). Closer to our proposal belongs AIT (Scherrer et al., 2021), which uses a neural network-based posterior model over the graphs but evaluates the F-score to select the interventions.

## 5 Experiments

We evaluate the performance of our method on synthetic and real-world causal experimental design problems and a range of baselines. We aim to investigate the following aspects empirically: (1) competitiveness of the overall proposed strategies of `Greedy-CBED` and `Soft-CBED` at scale (50 nodes) on synthetic datasets; (2) performance of the value acquisition strategy based on GP-UCB; and (3) performance of the proposed approach on a real-world inspired dataset.

### 5.1 Acquisition Functions

**Random.** Random baseline acquires interventional targets at random.

**AIT /** *soft***AIT.** active intervention targeting (AIT) (Scherrer et al., 2021) uses an f-score based acquisition strategy to select the intervention targets. See appendix B.6 for more details. Since the original proposed approach does not consider a batch setting, we introduce a variant that augments AIT with the proposed soft batching, as described in section 3.2.

**CBED / GreedyCBED / SoftCBED .** These are the Monte Carlo estimates of MI, as described in section 3. CBED selects a single intervention (target and value) that maximizes the MI and this intervention is applied for the whole batch. In `Greedy-CBED` , the batch is built up in a greedy fashion selecting the target, value pairs one at a time (Algorithm 1). `Soft-CBED` is sampling (target, value) pairs proportionally to the MI scores to select a batch, as described in section 3.2 and Algorithm 2.

## 5.2 Value Selection Strategies

**Fixed**: This value selection strategy assumes setting the value of the intervention to a fixed value. In the experiments, we fixed the value to $0$. **Sample-Dist**: This value selection strategy samples from the support of the observational data. **GP-UCB**: This strategy uses the proposed GP-UCB Bayesian optimization strategy to select the value that maximizes MI.

## 5.3 Tasks

**Synthetic Graphs.** In this setting, we generate Erdős–Rényi (Erdős and Rényi, 1959) (ER) and Scale-Free (SF) graphs (Barabási and Albert, 1999) of size 20 and 50. For linear SCMs, we sample the edge weights $\gamma$ uniformly at random. For the nonlinear SCM, we parameterize each variable to be a Gaussian whose mean is a nonlinear function of its parents. We model the nonlinear function with a neural network. In all settings, we set noise variance $\sigma^2 = 0.1$. For both types of graphs, we set the expected number of edges per vertex to 1. We provide more details about the experiments in appendix D.1.

**Single-Cell Protein-Signalling Network.** The DREAM family of benchmarks (Greenfield et al., 2010) are designed to evaluate causal discovery algorithms of the regulatory networks of a single cell. A set of ODEs and SDEs generates the dataset, simulating the reverse-engineered networks of single cells. We use `GeneNetWeaver` (Schaffter et al., 2011) to simulate the steady-state *wind-type* expression and single-gene *knockouts*. Refer to appendix D.2 for the exact settings.

## 5.4 Metrics

$\mathbb{E}$-**SHD**: Defined as the *expected structural hamming distance* between samples from the posterior model over graphs and the true graph $\mathbb{E}\text{-SHD} := \mathbb{E}_{\mathbf{g} \sim p(\mathcal{G}|\mathcal{D})}\big[\text{SHD}(\mathbf{g}, \tilde{\mathbf{g}})\big]$

$\mathbb{E}$-**SID**: As the SHD is agnostic to the notion of intervention, (Peters and Bühlmann, 2015) proposed the *expected structural interventional distance* ($\mathbb{E}$-**SID**) which quantifies the differences between graphs with respect to the causal inference statements and interventional distributions.

**AUROC**: The *area under the receiver operating characteristic curve* of the binary classification task of predicting the presence/ absence of all edges.

**AUPRC**: The *area under the precision-recall curve* of the binary classification task of predicting the presence/ absence of all edges.

## 6 Results

For each of the acquisition objectives and datasets, we present the mean and standard error of the expected structural hamming distance $\mathbb{E}$-**SHD**, expected structural interventional distance $\mathbb{E}$-**SID** (Peters and Bühlmann, 2015), area under the receiver operating characteristic curve **AUROC** and area under the precision-recall curve **AUPRC**. We evaluate these metrics as a function of the number of acquired interventional samples (or experiments), which helps quantitatively compare different acquisition strategies. Apart from $\mathbb{E}$-**SID**, we relegate results with other metrics to the appendix I.

On the synthetic graphs (Figure 3(a)), we can see that for ER and SF graphs with $50D$ variables and nonlinear functional relationships, the proposed approach based on soft top-k to select a batch with GP-UCB outperforms all the baselines in terms of the $\mathbb{E}$-**SID** metric. On the other hand, AIT alone does not converge to the ground truth graph fast even after combining with the proposed value acquisition, but when further augmented with the proposed soft strategy, the *soft*AIT recovers the ground truth causal graph upto 4 times faster and performs competitively to `Soft-CBED`. We observe similar performance across other metrics as well. In addition, we found this trend to hold for $20D$ variables and linear models. Full results are presented in the appendix I.

**Table 1:** Performance comparison between different value selection and batch strategies for `CBED`. Experiments are performed using an AMD EPYC 7662 64-Core CPU and Tesla V100 GPU.

| Strategy | | |
|---|---|---|
| **Value** | **Batch** | **Runtime(s)** |
| Fixed | Greedy | 32.56 |
| | Soft | 6.42 |
| GP-UCB | Greedy | 284.98 |
| | Soft | 24.17 |

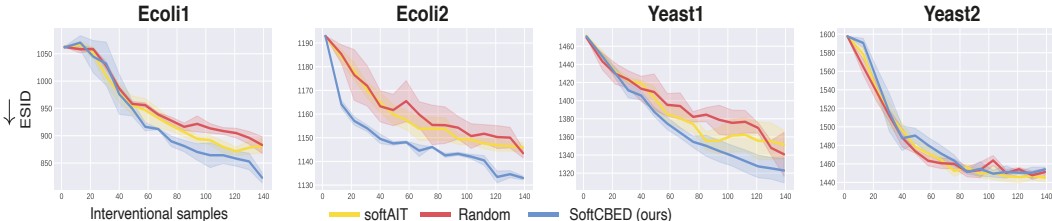

**Figure 4:** Comparison of acquisition functions on DREAM dataset, for 50 dimensions and batch size 10 on $\mathbb{E}$**-SID**↓ metric (6 seeds, with standard error of the mean).

Next, we examine the importance of having a value selection strategy for active causal discovery. We use the MI estimator in Equation 4; moreover, we test the proposed GP-UCB with two heuristics - the fixed value strategy and sampling values from the support. As we can see in Figure 3(b), selecting the value using GP-UCB clearly benefits the causal discovery process. We expect this finding as the mutual information is not constant with respect to the intervened value. To make this point clear, we demonstrate in the appendix G the influence of the value in a simple two variables graph. In addition, we note that naively sampling from the support of the observed dataset performs worse than fixing the value to 0. We hypothesize that this is due to lower epistemic uncertainty in the high density regions of the support, hinting that these regions might be less informative.

In order to further understand how the soft batch strategy compares with other batch selection strategies, we compare the results of `Soft-CBED` with `Greedy-CBED` and CBED. We observe (Figure 3(c)) that `Greedy-CBED` and `Soft-CBED` give very similar results overall. While `Greedy-CBED` is optimal under certain conditions (Kirsch et al., 2019), `Soft-CBED` remains competitive and has the advantage that the batch can be selected in a one-shot manner. This is also evident from the runtime performance of both these batching strategies in Table 1. Both these batch selection strategies perform significantly better than selecting one intervention target/value pair, and executing them $\mathcal{B}$ times (CBED).

Finally, on the DREAM task, we see that our method outperforms *soft*AIT and random baselines on the $\mathbb{E}$**-SID** metric (see Figure 4). In these experiments, since the intervention is emulating the gene knockout setting, we only use the fixed value strategy, with a value of $0.0$. Although random baseline still remains a competitive choice, in certain settings, `Soft-CBED` objective is significantly better (`Ecoli1`, `Ecoli2` datasets).

## 7    Summary and Conclusions

This paper studies the problem of efficiently selecting the Bayes optimal experiments to discover causal models. Our proposed framework simultaneously answers the questions of *where* and *how* to intervene in a batched setting. We present a Bayesian optimization strategy to acquire interventional targets and values. Further, we propose two different batching strategies: one based on greedy selection and the other based on soft top-k selection. The proposed methodology for selecting intervention target-value pairs in a batched setting provides superior performance over the state-of-the-art for causal models up to $50D$ variables. We validate this using synthetic datasets and using real-world inspired datasets of single-cell regulatory networks, showing the potential impact on areas like biology and other experimental sciences.

## Acknowledgments and Disclosure of Funding

We would like to thank Nino Scherrer, Tom Rainforth, Desi R. Ivanova and all anonymous reviewers for sharing their valuable feedback and insights. Panagiotis Tigas is supported by the UK EPSRC CDT in Autonomous Intelligent Machines and Systems (grant reference EP/L015897/1). We are grateful for compute from the Berzelius Cluster and the Swedish National Supercomputer Centre.

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
