# OpenReview forum: "Interventions, Where and How? Experimental Design for Causal Models at Scale"
_NeurIPS.cc/2022/Conference — NeurIPS 2022 Accept_

### Official Review · Reviewer_SUP9 · 2022-06-21

**Rating:** 7
**Confidence:** 2
**Soundness:** 4 excellent
**Presentation:** 3 good
**Contribution:** 4 excellent

**Summary:**

Causal discovery (CD) is an important field of research that can help find the causal relationship of the variables in a system. Only using observational data for finding causal structure, in the form of a Directed Acyclic Graph (DAG), is insufficient as there could be multiple DAGs that are Markov Equivalent. To resolve the uncertainties, interventional data can be incorporated to disambiguate between hypotheses. This work proposes a novel method based on Bayesian Optimal Experimental Design (BOED) that efficiently selects both intervention targets and values via the Bayesian optimization approach to find the true nonlinear structural causal model. Batch strategies, i.e., Greedy-CBED and Soft-CBED, are developed to select the most informative subset of interventions and values, as opposed to a single intervention at a time, to find the SCM.

Throughout the experiments, the authors demonstrate that their work can achieve better performance than existing methods on both synthetic (Erdos-Renyi, Scale Free) and real-world datasets (single-cell regulatory networks).


**Questions:**


i) Could the authors elaborate what is the newly introduced random variables $$ \Phi $$ in equation (3) when measuring the mutual information?

ii) What is the initialization strategy for the Gaussian Process model?

iii) What would happen when the number of variables in the SCMs is higher than 50?



**Limitations:**

Possibly the authors could provide some examples (such as the single-cell gene regulatory network) when the method fails to identify the true SCM.


**Strengths And Weaknesses:**


Strengths:
i) This paper is written exceptionally well; details of notations, definitions, equations, and derivation of theoretical results are provided.
ii) The proposed method is novel, as well as the proposed work, can find nonlinear SCMs and can select both intervention and values, in prior works are limited to linear models and can only select the intervention target.
iii) This work provides a comprehensive ablation study (Figure 3) so that we clearly better understand the effect of each adding component (such as the effect of selecting the intervention values, batch v.s. non-batch, and soft-CBED v.s Greedy CBED, on the E-SID performance. )

Weakness:

I do not particularly find a significant weakness in this work.

---

> ### Author Response · Authors · 2022-08-02
> **Response**
>
> We’d like to thank the reviewer for the positive comments regarding the writeup, novelty (*"The proposed method is novel, as well as the proposed work, can find nonlinear SCMs and can select both intervention and values, in prior works are limited to linear models and can only select the intervention target"*) and the evaluation protocol (*"comprehensive ablation study (Figure 3) so that we clearly better understand the effect of each adding component"*)
>
> **Regarding the weaknesses:**
>
> *“Could the authors elaborate what is the newly introduced random variables”*
>
> In line 108 we define lowercase $\phi$ as a composition of the parameters of the likelihood $\gamma$ and $\sigma^2$, and the graph $g$. The capital case $\Phi$ is defined as the random variable, consisting of the corresponding random variables over likelihood parameters and graphs, of which $\phi$ is a sample. We have modified the paper to fix this.
>
> “What is the initialization strategy for the Gaussian Process model”
> We’ve used the following GP hyperparameters and we have updated the appendix (E) to explicitly mention them.
> ```
> GP hyperparameters:
>     Matern kernel:
>         Length Scale 1.0
>         Length Scale bounds (lower=1e-5, upper=1e5)
>         Nu (smoothness of learned function) 2.5
>     Added 1e-6 to the diagonal of the kernel matrix
>     The number of restarts of the optimizer for finding the kernel’s parameters is 5
> ```
>
> *“What would happen when the number of variables in the SCMs is higher than 50”*
>
> This is a great question. The main bottleneck is not the posterior model DiBs, which is part of our current implementation but not part of our proposed method. One can change DiBs to a more optimized posterior model but we haven’t identified such a method yet.
>
> *“Possibly the authors could provide some examples (such as the single-cell gene regulatory network) when the method fails to identify the true SCM.”*
>
> This is an excellent question. In real-world single-cell gene regulatory networks like the real-world inspired DREAM, the underlying causal process can be circular and the in-silico simulators have proposed a DAG approximation to make the graph identifiable (or *more identifiable* since there are no guarantees in the general setting, without additional assumptions like additive gaussian noise), for the needs of the benchmark. We will update the limitations to mention that our method is limited to DAG approximated gene regulatory networks and it cannot work in the more general complex feedback and circular causality GRNs case.

---

> > ### Comment · Reviewer_SUP9 · 2022-08-08
> > **Thank you.**
> >
> > I sincerely thank the authors for the clarification of the notation and the use of the GP model. This significantly answers my questions. I maintain my current score.

---

### Official Review · Reviewer_L9rC · 2022-07-05

**Rating:** 5
**Confidence:** 3
**Soundness:** 2 fair
**Presentation:** 2 fair
**Contribution:** 3 good

**Summary:**

This paper proposes to solve the problem of causal discovery through an experimental design method which selects interventions based on the mutual information between an intervention and the SCM. As evaluating the mutual information in closed form is not possible, the authors propose to use a BO algorithm to identify its maximum which corresponds to the experiment that should be evaluated next. The authors additionally extend the method to cases in which the experimenter wishes to select a batch of data points.

**Questions:**

See section above.

**Limitations:**

Yes, this is discussed in the supplement. I cannot identify additional potential negative societal impact for this work.







**Strengths And Weaknesses:**

Overall I think the problem the authors are addressing is important. However, in terms of novelty, this paper seems to be an extended and detailed version of [42] where all the main ideas were already presented. One important difference is in the selection of the intervention value together with the intervention set. It would be great if the authors could provide a list of contributions to highlight the novelty of the paper. A list of detailed comments and questions is given below.

**Background section.** Even though I am very familiar with the topic discussed and the notation generally used in causality, I found the background section difficult. I believe this is due to the missing definitions and typos.
- Line 106-108 is confusing because of the typos. Do you need a $i$ for $f$, $\gamma$ and $\sigma$? Is $\theta$ a collection of these parameters for all $i$’s?
- In general, causal discovery refers to the problem of learning the DAG and not the functional relationships. While both are connected in this paper you seem to be addressing these two problems jointly. Is my understanding correct? Can you clarify this point?
- The notation is a bit imprecise and a few things are not defined in section 2. For instance E at line 85. In addition, you need to define what you mean with the do operator and what $x_w$ is. What are $\phi$ and $\Phi$ at line 127? The parameters of the posterior distribution on the SCM (which was also not introduced)?
- What is VCN?
- First formula after line 89, typos in the parenthesis I guess it should be $do(X_w = x_w)$. Same for parenthesis in Eq 1.
- Add at least one reference for the background section, for instance when you define SCM. SCM generally refers to $(V, U, F, P(U))$ (U are the exogenous variables). The SCM is associated with a DAG.
- Line 106 -> $f_i$
- Provide at least one reference for the definition of mutual information. It seems to me that the first terms should only be conditioned on D.

**Method section.**
- Clarify if it is possible to observe all variables after implementing an intervention. Is $Y = V\backslash X_j$?
- How is the prior on $\Phi$ defined? This is the joint prior distribution on the graph and the parameters of the functional relationships and the noise variances? This is never explicitly stated.
- Even though there exists a discrete set of intervention variables that we can manipulate (especially given Assumption 4) the number could be very large depending on the size of the graph. I don’t see how this makes the method scalable.
- I think you should compare finding the optimum of the MI with BO with a grid approach that finds the optimum by numerically evaluating the function MI for every j on a set of points. I would be interested in seeing what is the reduction in terms of function evaluations that you get with BO.


**General comments.**
- Assumption 4. With target to you refer to the intervened variable? Where is this assumption used? What is preventing you from doing interventions on more than one variable? Is it cause exploring interventions on all subset of variables would make the approach less scalable?
- Assumption 3. Where is this assumption used?  Is this assumed to be able to write the likelihood?
In general it is not clear why and where these assumptions are needed.
- The authors speak about causal discovery at scale. However, the approach presented requires enumerating all possible DAGs on which the prior needs to be defined. How is this problem addressed?
- I guess selecting the intervention level is relevant when the intervened variables are continuous. However this is not stated anywhere in the paper.
- I am confused about the use of observational and interventional data. While the observational data are used to compute (4) (via DiBS?) how are the interventional data collected sequentially used? Are these just included in $\mathcal{D}$? The authors should clarify how exactly $\mathcal{D}$ is used in the computation of (4) and how the interventional data are treated.
- Are Alg 1 and 2 just giving one step of the overall algorithm? For how many steps is this repeated? I could not find this mentioned for the experiments presented.
- Regarding scalability, as mentioned above, it would be great if the authors could clarify what makes this method scalable with respect to the existing ones. The fact that the experiments are on graphs with a large number of nodes is great but is the capability of dealing with a large number of nodes a feature of the method or of the implementation?

---

> ### Author Response · Authors · 2022-08-02
> **Partial Response 2**
>
> We’d like to thank the reviewer for acknowledging the fact that the problem we are addressing is important. Also, we appreciate the attention to detail. We address the comments below:
>
> *“Novelty with respect to [42]”*
>
> Although [42]([49] in the revision) is an inspiration to our method, we are different on a couple of points.
> [42 / 49 in revision] addresses the problem of Bayesian causal experimental design, under the modelling assumption that the graphs can be enumerated and the likelihoods are modelled as Gaussian Processes. As the number of nodes increases, the enumeration becomes prohibitive and as such, the problem needs a significantly different approach. We employ the use of an approximate posterior model (DiBs) which allows for sampling from the posterior $p(\Phi \mid D)$ which alleviates the limitation from [42 / 49 in revision] (the authors demonstrate the method only on two node setting).
> However, by using DiBS instead of GP/ enumerating the graphs, eq 6 from [42 / 49 in revision] is not applicable anymore for two reasons: 1. We have access to only samples from the posterior and as such we have to rely on Monte Carlo estimators and 2. We can’t evaluate the entropy over parameters (LHS integral) without updating the posterior model, which in our case will require solving an expensive optimization problem). For this, we need to use the Bayesian Optimal Experimental Design objective which evaluates the entropies over the outcomes (eq 4 in our paper)  – a quantity which is easier to compute via the use of Monte Carlo approximation and has been used in Deep Active Learning in an objective also known as the BALD objective. In short, we employ a series of components that are scalable to solve SCMs of a high number of nodes, a limitation that [42 / 49 in revision] cannot address.
>
> *“Line 106-108 is confusing because of the typos. “*
>
> The parameters indeed contain the likelihood parametrizations per node and the global graph $g$. We have omitted the subscripts to simplify the notation.
>
> *“In general, causal discovery refers to the problem of learning the DAG and not the functional relationships..”*
>
> This is correct, we are addressing these two jointly however this should lower bound the performance of our method since it might “waste” acquisitions on experiments of high expected information gain due to the uncertainty over likelihoods parameters. A way to address this would be to marginalize over the model parameters, however DiBs, as a method, samples graphs and likelihoods parameters pairs, so we eventually acquire experiments using the Expected Information Gain (EIG) over the joint. We’d like to point out that our evaluation is focused on causal discovery (thus the SHD, SID, AUROC, AURPC metrics over the adjacency matrices) and we show that even though we acquire experiments based on the EIG over both DAGs and functional relationships (the likelihoods) we outperform the existing baselines.
>
> *“The notation is a bit imprecise and a few things are not defined in section 2”*
>
> $\phi$ refers to a sample from a distribution defined over the random variable $\Phi$. The parameters of the posterior over the SCMs is mentioned in 127 (we have fixed a typo on the formula below where q_\phi should have been q_\psi.
>
> *“What is VCN?”*
>
> Variational Causal Networks - We have updated the text with the full name.
>
> *“Typos … First formula after line 89,”*
>
> Correct - thank you for your attention to those details. We have fixed those issues on the updated manuscript.
>
> *“Add at least one reference for the background section, for instance when you define SCM. SCM generally refers to (V,U,F,P(U))  (U are the exogenous variables)...“*
>
> Our definition should be compatible with the (V,U,F,P(U)) definition you suggest. Specifically, eq. 2 describes the endogenous variables, exogenous variables, functional dependencies and the distribution over the exogenous variables. We have added as a reference Peters et al., Elements of Causal inference, 2017 and Judea Pearl Causality 2009.
>
> *“Line 106 -> f_i”*
>
> We omit the subscript to cover all the functional relationships and simplify the notation. We have updated the text to clarify this.
>
> *“Provide at least one reference for the definition of mutual information”*
>
> The common formulation in Bayesian Optimal Experimental Design/ Bayesian Active Learning is to condition on the data. However, in formula under 139, we should have had $p(\theta \mid D)$ instead of $p(\theta)$ and we’ll correct this. Also, we’ll add the following references:
> - [1] Bayesian Active Learning for Classification and Preference Learning
> - [2] Deep Bayesian Active Learning with Image Data
>
> *“Clarify if it is possible to observe all variables after implementing an intervention”*
>
> This is correct and it’s implied by the causal sufficiency assumption (assumption 1).
>
> *“How is the prior on $\Phi$  defined? “*
>
> We omit the details of how $\Phi$ is defined because it’s discussed in the DiBs paper in detail.

---

> ### Author Response · Authors · 2022-08-02
> **Partial Response 1**
>
> *“Even though there exists a discrete set of intervention variables that we can manipulate (especially given Assumption 4) the number could be very large depending on the size of the graph...”*
>
> This is a great question. A large number of nodes will increase the number of parallel Bayesian Optimization (BO) runs we’ll run (one per node). Although it's true that this can become a problem on very large graphs, in practice, we observed that updating the posterior model is the bottleneck and the parallel run of the BO was lightweight enough to be a problem.
>
> *“I think you should compare finding the optimum of the MI with BO with a grid approach that finds the optimum by numerically evaluating the function MI for every j on a set of points...”*
>
> This is an excellent remark, however, grid search is expected to be very sample inefficient as compared to BO. While such an experiment would demonstrate the performance of BO vs grid search, we believe such an ablation is out of scope of our evaluation protocol.
>
> *“Assumption 4. With target to you refer to the intervened variable? Where is this assumption used? “*
>
> By intervened variable we refer to the target of the experiment or the intervention and the assumption is used in our method (we don’t allow multi-target/parallel interventions). Extending our setting to multi-target while selecting the value as well would make the problem significantly more challenging (combinatorial explosion), thus we are focused on the single (atomic) target setting. We leave the multi-target (and admittedly more realistic but also challenging setting) for future work.
>
> *“Assumption 3. Where is this assumption used?”*
>
> Non-linear assumption extends the class of SCMs that our method can address. This is implicitly used by DiBs (which support non-linear likelihoods). The gaussian additive noise is a common assumption in causal discovery since without such assumptions the true SCMs might not be identifiable (Peters et al, "Causal Discovery with Continuous Additive Noise Models")
>
> *“The approach presented requires enumerating all possible DAGs on which the prior needs to be defined...“*
>
> Our method **does not** require enumerating all possible DAGs. This is because the posterior model we use (DiBs), allows for sampling posterior samples through the use of Stein Variational Gradient Descent. Thus when computing the expected information gain, we can employ a monte carlo integration method to approximate it (eq. 8) with the use only of the posterior samples (instead of enumerating all the graphs).
>
> *“I guess selecting the intervention level is relevant when the intervened variables are continuous...“*
>
> That’s correct and we’ll update the manuscript to be explicit about it. However, we’d like to emphasize that continuous variable setting is more challenging, and in the discrete case, a grid search might be sufficient (assuming a small number of discrete values).
>
> *“I am confused about the use of observational and interventional data...“*
>
> in (4), the D is used for the computation of $p(\Phi \mid D)$. The dataset $D$ contains a concatenation of both observational and interventional data. DiBs belongs to the score-based causal discovery methods where the score consists of a likelihood term (probability of observations or interventions given the parameters of the SCM) and a regularization. For the interventional setting, we truncate the likelihood to exclude the outcomes that are intervened (instead of sampled). We will clarify this in the appendix.
>
> *“Are Alg 1 and 2 just giving one step of the overall algorithm? For how many steps is this repeated?”*
>
> Algorithms 1 and 2 refer to the acquisition of a single batch. D.2 contains the number of iterations (steps) and the size of each batch for the DREAM dataset. For the synthetic experiments we used batch size 10 and num batches 10, we’ve updated the appendix to include this.
>
> *“Regarding scalability, as mentioned above, it would be great if the authors could clarify what makes this method scalable with respect to the existing ones...“*
>
> The novelty of our method is that we scale up batch causal discovery to the less restrictive class of non-linear SCMs of large variables with the help of monte carlo estimators (avoiding the enumeration of the graphs via the use of samples from the approximate posterior), the bayesian optimization over the values and the stochastic batch selection. The scalability of the method thus becomes a feature of the combination of all these ingredients which compose the method and not the individual implementation (i.e. DiBs will not make an active causal discovery method scalable without the use of monte carlo estimators and stochastic selection of the batch).

---

### Official Review · Reviewer_25Ya · 2022-07-07

**Rating:** 8
**Confidence:** 3
**Soundness:** 4 excellent
**Presentation:** 4 excellent
**Contribution:** 3 good

**Summary:**

This paper considers Bayesian optimal intervention selection for active causal discovery. The problem setting is causally sufficient causal graphs under additive noise models and atomic interventions. Importantly, both the target nodes to intervene and intervention values that will be assigned to the target nodes are optimized in a batched setting. Mutual information between the observations and the parameters is optimized in Bayesian Optimal Experimental Design context. Variational inference is used to approximate the posterior over SCMs. After thorough derivations, Monte Carlo approximations are used to estimate MI for single intervention design. For selecting the target value over a continuous domain, a Gaussian Process is used along with Upper Confidence Bound mechanism. Furthermore, two approaches (greedy and top-k selection) are proposed for batch design. Experiments on synthetic datasets demonstrate the effectiveness of each piece of the proposed method. Experiments on semi-real datasets also show promising results.

**Questions:**

Regarding the weakness above, do you see an extension to non-atomic interventions possible within the proposed framework?

On a smaller note, L53-54 states: “existing work on active causal discovery focuses exclusively on selecting the intervention target”. Although it solves a different problem, "Matching a Desired Causal State via Shift Interventions" (Zhang et al. 2021 NeurIPS) can be a relevant work that also aims to find the intervention values.

**Limitations:**

Assumptions of the paper, e.g. Gaussian noise and atomic interventions, are clearly stated in the paper. Perhaps the second point can be addressed in a bit more detail.

**Strengths And Weaknesses:**

The paper is well-written, with a sufficient amount of details in the main body and a lot of secondary information on the appendices. The proposed method consists of many components, some borrowed from the existing literature such as GP-UCB and DiBS posterior model. The paper does a nice job putting them all together coherently and explaining the steps.

The considered problem is general, and I believe the main contribution is this generality: allowing non-linear systems, scaling up to 50 nodes (though in quite sparse graphs), and the ability to set the intervention values rather than just finding targets.

Especially, the last point seems important. Although there are other information-theoretic approaches to causal discovery, to my knowledge, the present paper is the first one that studies the effect of the target value and supports its claims through extensive experiments.

The only major weakness is the assumption of atomic interventions. For instance, ABCD (Agrawal et al., 2019) allows multiple nodes to be intervened simultaneously. Although the proposed batch solutions are helpful for faster causal discovery, the assumption of atomic interventions can be a limiting factor.

---

> ### Author Response · Authors · 2022-08-02
> **Response**
>
> We would like to thank the reviewer for acknowledging the exposition of our paper (*“the paper is well written”*) and the novelty of our method (*“The proposed method consists of many components, some borrowed from the existing literature such as GP-UCB and DiBS posterior model. The paper does a nice job putting them all together coherently and explaining the steps” and “the present paper is the first one that studies the effect of the target value and supports its claims through extensive experiments”*)
>
> **Regarding the weaknesses:**
>
> We understand that atomic interventions is a limiting assumption in our work. However, dealing with non-atomic batch interventions while optimizing for the value as well makes the problem significantly more challenging (combinatorial explosion). We’d like to stress out though that the expected information gain is not limited to atomic interventions. Similarly to ABCD, we can adjust the conditional distributions and the approximate posterior model (ingredients of the EIG)  to support multi-target interventions, however, by doing so, like ABCD, we will need to enumerate the set of all possible interventions to construct the interventional set, which will grow exponentially to the number of nodes.  We aim to address this limitation in future work, however for now we want to emphasize that selecting both the node and the value of the intervention is a very important aspect of causal discovery methods albeit not suitable for scalable multitarget interventions.
>
> *“Although it solves a different problem, "Matching a Desired Causal State via Shift Interventions" (Zhang et al. 2021 NeurIPS) can be a relevant work that also aims to find the intervention values.“*
>
> Thank you for the reference. It is true that this is a slightly different setting but we have added this to the related work.

---

### Official Review · Reviewer_3PzT · 2022-07-10

**Rating:** 5
**Confidence:** 3
**Soundness:** 3 good
**Presentation:** 3 good
**Contribution:** 2 fair

**Summary:**

The paper proposes a Bayesian approach to learning causal relationships from a mix of observational and interventional data. Compared to existing approaches, the proposed method can cope with the nonlinearity of the causal relationships and can select both target and value to intervene with to facilitate more efficient learning.

**Questions:**

I don't have additional questions other than my concerns expressed in the weakness section.

**Limitations:**

It is not clear whether the authors have sufficiently discussed the limitations of the paper. Assumptions such as causal sufficiency may be considered as limitations.

**Strengths And Weaknesses:**

Strengths:
* The paper tackles a long-standing problem in causal discovery. Learning from a mix of observational and interventional data is an interesting and relevant setting in causal inference and discovery.
* The proposed approach is reasonable and is capable of handling a few challenges as mentioned in the paper including nonlinearity and choosing both the target and value to intervene.
* The paper is well written and easy to follow.

Weakness:
* The proposed method is somewhat non-surprising and there is a lack of formal (theoretical) justification of the performance of the proposed method.
* While experiments on both synthetic and real-world data demonstrate the performance of the proposed method compared to alternatives. It is not clear to me whether the proposed method has been compared to sufficient and competitive enough alternatives. The use of only one real-world data, while understandable, is also somewhat restrictive. The paper can also benefit from a robustness study evaluating when the key assumptions in the paper are violated.

---

> ### Author Response · Authors · 2022-08-02
> **Response**
>
> We would like to thank the reviewer for acknowledging the exposition of our paper (*“The paper is well written and easy to follow.”*), the importance of our setting (*"The paper tackles a long-standing problem in causal discovery. Learning from a mix of observational and interventional data is an interesting and relevant setting in causal inference and discovery."*) and emphasizing the strengths of our method (*"The proposed approach is reasonable and is capable of handling a few challenges as mentioned in the paper including nonlinearity and choosing both the target and value to intervene."*).
>
> **Regarding the weaknesses:**
>
> *“The proposed method is somewhat non-surprising and there is a lack of formal (theoretical) justification of the performance of the proposed method.”*
>
> Please note that the proposed greedy strategy is (1-1/e) optimal due to submodularity (which is proved in the appendix). For our stochastic batch selection strategy the regret bound is left for future work as the accumulation of the regret bounds and stochastic sampling makes the analysis non-trivial.
>
> *“It is not clear to me whether the proposed method has been compared to sufficient and competitive enough alternatives.”*
>
> Can you please point us to relevant baselines in *Bayesian experimental design with the same assumptions* (arbitrary graphs without assuming access to skeleton, nonlinear SCM and no hidden confounders) which we might have missed? To the best of our knowledge, we have compared all the relevant and competitive Bayesian Experimental Design for Causal Discovery methods (ABCD, AIT). But we are happy to compare with other relevant baselines if they exist.
>
> *“The paper can also benefit from a robustness study evaluating when the key assumptions in the paper are violated.”*
>
> Unfortunately, evaluating the experimental design strategy when these assumptions are not satisfied is beyond the scope of the current work as this would entail introducing different posterior models and an overall acquisition strategy. See the comment under the response to all the reviewers for a detailed explanation.

---

### Official Review · Reviewer_PxXf · 2022-07-12

**Rating:** 6
**Confidence:** 3
**Soundness:** 3 good
**Presentation:** 2 fair
**Contribution:** 2 fair

**Summary:**

The paper presents a bayesian intervention design method for single-node interventions, including the target values, in causal models with additive noise.

**Questions:**

Some of the related work https://papers.nips.cc/paper/2019/hash/5ee5605917626676f6a285fa4c10f7b0-Abstract.html shows that maximising the expected information gain might not be the optimal strategy in their setting, so I was curious about how this would apply in the setting of this paper

In general, I would be curious about how this work compares to all of the missing related work.

**Limitations:**

The paper proposes an interesting approach, although limited to single-node interventions and ANMs (which seems a strong assumptions). One big limitation in my opinion is that it fails to cite most of the related work.

**Strengths And Weaknesses:**

*** EDIT after rebuttal *** I think the authors did a great job in the rebuttal and answered most of my (and other reviewers') concerns (including the similarity with previous work on GPs for BOED [49]), so I changed my score to weak accept.

Strengths:
- Intervention design is an important, and in my opinion, understudied problem
- The paper presents an interesting and to the best of my knowledge theoretically sound method

Weaknesses:
- The paper seems to ignore a lot of related work on intervention design, mostly focusing on differential causal discovery. This is a bit weird since several of these these papers are cited in [1] and [38] and the whole terminology of this paper is based on related work that isn't cited, e.g. intervention design with a variety of settings (active/passive, multiple targets, budgeted, latent variables and also Bayesian): https://proceedings.neurips.cc/paper/2015/hash/b865367fc4c0845c0682bd466e6ebf4c-Abstract.html, https://proceedings.neurips.cc/paper/2020/hash/f57bd0a58e953e5c43cd4a4e5af46138-Abstract.html, https://proceedings.mlr.press/v70/kocaoglu17a.html, https://proceedings.neurips.cc/paper/2017/hash/291d43c696d8c3704cdbe0a72ade5f6c-Abstract.html, https://proceedings.neurips.cc/paper/2021/hash/0b94ce08688c6389ce7b68c52ce3f8c7-Abstract.html, https://papers.nips.cc/paper/2019/hash/5ee5605917626676f6a285fa4c10f7b0-Abstract.html  and many more

- It seems to be also missing another even more related work on optimal experiment design with GPs: https://openreview.net/forum?id=Ih_ogoAw5G

---

> ### Author Response · Authors · 2022-08-02
> **Response**
>
> We thank the reviewer for their valuable feedback.
>
> We’d like to point out that https://openreview.net/forum?id=Ih_ogoAw5G has already been cited as [42]([49] in revision) and discussed in the related work section. Please also see below the response to reviewer L9rC for the details on how our work differs from this work.
>
> Indeed, we have focused the related work on the differential causal discovery under the presence of a finite set of observational and interventional data, where a Bayesian Optimal Experimental Design approach becomes a theoretically justified method.  We have hence discussed in our related work prior works which have addressed BOED for causal discovery under a similar set of assumptions. We highlight how the papers highlighted differ from our work:
>
> [1] https://proceedings.neurips.cc/paper/2015/hash/b865367fc4c0845c0682bd466e6ebf4c-Abstract.html,
> [2] https://proceedings.neurips.cc/paper/2020/hash/f57bd0a58e953e5c43cd4a4e5af46138-Abstract.html and
> [3] https://proceedings.mlr.press/v70/kocaoglu17a.html are not based on Bayesian Optimal Experimental Design and also assume that a skeleton/ a partially directed graph is available initially. These works deal with active learning only the intervention targets to orient the edges of the PDAG  and theoretically analyse the performance of the proposed method based on separating systems, clique trees and when each intervention has a cost associated with it respectively. This is different from our work which is based on BOECD.  We also do not assume access to any skeleton/ PDAG initially. In addition, besides just selecting the intervention target, we also select the intervention value. As these papers are not based on Bayesian approach to experimental design, we have not discussed them. But we have added them in the related work in the light that still deals with active learning/ experimental design for causality under different assumptions than our work.
>
> [4] https://proceedings.neurips.cc/paper/2017/hash/291d43c696d8c3704cdbe0a72ade5f6c-Abstract.html proposes a two-   approach for causal experimental design in the presence of latent variables (thereby not assuming causal sufficiency). While this is an interesting problem, it is different from our work in that we assume causal sufficiency and tackle the problem using BOED allowing for nonlinear models and strong empirical performance at scale. We agree it might be relevant in some applications to relax the causal sufficiency assumption, and we have discussed this aspect in length in the comments to “response to all the reviewers”, where we highlight BOED with causal sufficiency assumption.
>
> [5] https://proceedings.neurips.cc/paper/2021/hash/0b94ce08688c6389ce7b68c52ce3f8c7-Abstract.html is concerned with estimating the unknown intervention target from observational and interventional data and not experimental design. As such, this work deals with a fundamentally different problem from what we consider.
>
> [6] https://papers.nips.cc/paper/2019/hash/5ee5605917626676f6a285fa4c10f7b0-Abstract.html Though this work is based on Bayesian approach to experimental design similar to our work, the above paper assumes that each of the undirected components of the essential graph is always a tree. Without further domain knowledge, this assumption might be a bit restrictive in many applications. On the other hand, we are interested in the general DAG setting (graph agnostic thus we can't exploit any graph structure). As discussed in our work, in a general DAG setting, maximising EIG is optimal but it need not necessarily be the case if certain graph-related inductive biases can be exploited. We have updated the related work to include this paper.
>
> We would be happy to answer any other questions/ clarifications you might have regarding the work. Please let us know. If our response satisfactorily answered all your questions, we would appreciate it if you could consider updating the rating.

---

> > ### Comment · Reviewer_PxXf · 2022-08-08
> > **Thanks for the rebuttal, I changed my score to weak accept**
> >
> > I think the authors did a great job in the rebuttal and answered most of my (and other reviewers') concerns (including the similarity with previous work on GPs for BOED [49]), so I changed my score to weak accept.

---

### Official Review · Reviewer_h91K · 2022-07-17

**Rating:** 6
**Confidence:** 4
**Soundness:** 3 good
**Presentation:** 3 good
**Contribution:** 2 fair

**Summary:**

The authors proposed to efficiently learn a causal structure through interventions under a causal sufficiency assumption. The key idea is how to design experiments (intervention) in a way to figure out the causal structure where they adopted Bayesian in that (i) use of an (approximate) posterior distribution over candidate graph structures and parameters and (ii) (approximate) Bayesian Optimal Experimental Design in performing experiments. More specifically, what variable and which value to intervene to get a sample is determined with respect to maximizing mutual information between the parameters (structural and functional) and observation, where they adopted Bayesian Optimization. Single and Batch designs are proposed with two options for Batch, Greedy and Softmax-based.


**Questions:**

- Please define beta in Line 193-4 in the main text.
- Gaussian noise is assumed, but it is unclear where the assumption is explicitly used. It doesn’t seem to be related to Line 188-189
- What is the main novelty of the *-CBED? Also, it would be nice if you could emphasize the contributions of the paper?


**Ethics Review Area:**

["I don’t know"]

**Limitations:**

- Causal sufficiency assumption and that all variables are intervenable. These two guarantees identifying a causal graph.
- Only one variable is intervenable at a time.
- Assumption 4 prohibits the use of discrete variables.


**Strengths And Weaknesses:**

Strengths
- The paper focuses on scaling up Bayesian causal discovery, and there are several places where the authors tried to incorporate recent works to approximate intractable quantities.
- The efficiency of the algorithm is demonstrated empirically.

Weaknesses
- The main idea of the paper looks like a collection of existing methods (BALD, DiBS, BO, Soft Top-K). Surely, putting them into a piece neatly to solve a problem would require expert knowledge of the ingredients.
- Tests on a synthetic graph, where the expected number of edges per vertex is one, will likely result in multiple independent components. Experiments on small/medium size graphs with a bit more dense edges are desired. Furthermore, tests under unobserved confounders (i.e., violation of assumptions) would be a nice addition.
- Noise variance seems small (but is an understandable choice given that it would not likely affect comparisons among methods).
- The authors misused the term SCM in many places. SCM consists of functions, and it induces a causal graph (not part of it). The graph can be identified, and parameters can be learned, but SCM won’t be typically identified. (Line 66, recovering the original SCM, Line 44 consists of a DAG structure … )

---

> ### Author Response · Authors · 2022-08-02
> **Response**
>
> We thank the reviewer for the valuable feedback. We address each of your concerns and questions below:
>
> *Define beta in 193-4*
>
> $\beta$ is the hyperparameter of GP-UCB which balances exploration and exploitation. We have updated the paper with this clarification.
>
> *Place where Gaussian noise assumption is used*
>
> The Gaussian noise assumption is needed because the underlying posterior model that we employ, DiBS, makes this assumption. But from the perspective of our proposed experimental design CBED framework, a Gaussian noise assumption is not necessary and the proposed acquisition strategy is still valid without it. Hence, Gaussian noise assumption can be relaxed if a different posterior model which does not make this assumption is used.
>
> *On experiments with sparse graphs*
>
> We have mainly focused on the sparse graphs setting for our experiments because the causal graphs found in real world are sparse (See Bengio 2019 and Schmidt et al 2007 for a detailed discussion on this). We also found that in the DREAM dataset which contains real world inspired causal graphs, the graphs on average have an expected number of edges between 1 and 1.5, similar to our synthetic benchmark setting. In addition, sparsity influences the ratio between informative targets and uninformative targets. Denser graphs have more number of informative targets as many nodes are likely to have causal parents while sparse graphs will have fewer informative targets. Therefore, in denser graphs, a random policy might be sufficient as such a policy is very likely to select an informative target. So an experimental design strategy is more desirable in a sparse graph setting than a denser one.
>
> *Main novelty of CBED*
>
>  The main novelty of the proposed CBED framework is a working method for experimental design for the problem of causal discovery. With an efficient approach and thorough empirical evaluation, we show that acquiring values along with intervention targets in a batch setting is important and can be helpful in recovering the causal graphs faster. As reviewer 25Ya indicated, this has not been demonstrated before in experimental design causal discovery. While it is true that some of the components used in the algorithm already exist (like BO, DibS), the contribution of this paper is to demonstrate to the community a principled methodology for acquiring both interventional targets and values using these components (and this requires expert knowledge as you indicate, and cannot be achieved by just combining the existing components). As such, our contribution should not be seen as just a synthesis of these components and we argue that the novelty comes from the overall method for this specific problem.
>
> *Noise variance seems small (but is an understandable choice given that it would not likely affect comparisons among methods).*
>
> We agree that the variance might seem small however as you pointed out, this setting is the same among all the baselines thus the lower bound of the sample efficiency should be the same.
>
> *Regarding causal sufficiency*
>
> See note to all reviewers above.
>
> *Regarding all variables are intervenable*
>
> We agree that in some applications not all variables are intervenable. An easy fix to this is to only consider intervenable nodes when optimizing the expected information gain. Causal sufficiency on the other hand is significantly more challenging to relax (see causal sufficiency comment in the "response to all authors"). We will discuss this limitation and surface it in the extra page of the camera ready.
>
> *Only one variable intervenable at a time*
>
> See note to all reviewers above.
>
> *Assumption 4 prohibits discrete variables*
>
> We assume you meant to say assumption 3 (Gaussian additive noise) prohibits the use of discrete variables. As outlined before, it is an assumption required for the posterior model DiBS, but is not necessary for our proposed experimental design strategy. Hence, CBED can be used when discrete variables are present as well.
>
> *Regarding misused the term SCM*
>
> Thanks for pointing this out. We have made the correction in the revision.
>
> We would be happy to answer any other questions/ clarifications you might have regarding the work. Please let us know. If our response satisfactorily answered all your questions, we would appreciate it if you could consider updating the rating.

---

> > ### Comment · Reviewer_h91K · 2022-08-08
> > **..**
> >
> > Thanks for the clarification. Regarding the explanation (or conjecture?) for the "denser graphs" and "random policy", I would like to see it incorporated in the main paper. Thanks again!

---

### Author Response · Authors · 2022-08-02
**To All Reviewers**

We thank all the reviewers for the detailed feedback. We are pleased that the reviewers find that we explore an “important” [L9rC] problem and “present an interesting and theoretically sound method”[PxXf]. Furthermore, they find our contribution is “novel”, “can select both intervention and values” and “ is the first one that studies the effect of the target value and supports its claims through extensive experiments [SUP9, 25Ya]. Finally, they find that our work is *“written exceptionally well”* and *“easy to follow.”* [SUP9, 3PzT].

We would like to clarify some common comments reviewers have mentioned regarding the assumptions:

*On Causal sufficiency*

This is a common assumption in the current causal discovery literature (see [1,2,6,9,18,26] for instance). Relaxing this assumption or evaluating the proposed experimental design method when this assumption is not satisfied would involve significant challenges which go beyond the scope of the current work. An additional layer of complexity arises when experimental design is also involved. Besides, we are not aware of any posterior models (i.e. Bayesian Causal structure learning frameworks) which can handle hidden confounders seamlessly at scale, thereby limiting our capabilities on what we can explore in the current work. However, we agree that studying causal discovery under hidden confounders is important and we aim to investigate this as part of our future work. We will highlight this point in the limitations and surface it on the additional page of the camera ready.

*On assumption of atomic interventions*

Reviewers 25Ya,h91K commented on the possibility of multi-target experimental design and the limitation that the assumption of atomic interventions brings forth. We agree that multi-target interventions could be important in some settings. However, while the multi-target experimental design is relevant, the problem of *optimally* or *near-optimally* selecting *multi-target and value pairs* simultaneously is a combinatorial problem and is very hard. This is due to the fact that the target set will become a powerset over the nodes and this will increase the number of Bayesian Optimization runs exponentially to the number of nodes. We leave this for future work but will highlight this aspect in limitations and surface it on the additional page of the camera ready.

*On assumption of additive Gaussian noise variables*

The assumption of additive noise variables which are also Gaussian is required only because the underlying posterior model that we use (DiBS) makes this assumption. But from our proposed  BOED methodology perspective, these assumptions are not necessary and can be potentially relaxed with a different posterior model that does not require these assumptions. Note that this also means that if the variables were non-Gaussian or discrete, our experimental design strategy would still apply. For the discrete variable case, a grid search to select the intervention values can be performed instead of GP-UCB.

As a final point, we highlight that we relax other assumptions like allowing non-linearity of the underlying SCM while allowing for the batch acquisition of intervening target-value pairs.

---

### Author Response · Authors · 2022-08-07
**gentle reminder**

Dear reviewers,

As the discussion period is ending soon, we’d like to ask you to discuss any other concerns regarding our work, or if you’re happy with our response, please consider updating the score. We are happy to address any open questions you have which might help in raising the score.

---

### Meta-Review · Area_Chair_1o1y · 2022-08-26

**Recommendation:** Accept
**Confidence:** Certain

**Metareview:**

The paper proposes a Bayesian optimization strategy for causal discovery under causal sufficiency and additive noise. The main point is to choose interventions that maximize mutual information between parameters and observations. The procedure combines techniques from several existing methods. The authors have successfully addressed questions raised by reviewer PxXf about related work. Several reviewers praise the writing (SUP9,3PzT,25Ya). Overall, this is a strong paper, with atomic interventions as the main limitation (see reviewers 25Ya,h91K)

**Award:**

No

---

### Decision · Program_Chairs · 2022-09-14

Accept